# REVISITING HIGH-RESOLUTION ODES FOR FASTER CONVERGENCE RATES

## ABSTRACT

There has been a growing interest in high-resolution ordinary differential equations (HR-ODEs) for investigating the dynamics and convergence characteristics of momentum-based optimization algorithms. As a result, the literature contains a number of HR-ODEs that represent diverse methods. In this work, we demonstrate that these different HR-ODEs can be unified as special cases of a general HR-ODE model with varying parameters. In addition, by using the integral quadratic constraints from robust control theory, we introduce a general Lyapunov function for the convergence analysis of the proposed HR-ODE. Not only can a large number of popular optimization algorithms be viewed as discretizations of our general HR-ODE, but our analysis also leads to several critical improvements in the convergence guarantees of these methods both in continuous and discrete-time settings. The notable improvements include enhanced convergence guarantees, compared to prior art, for the triple momentum method ODE in continuous-time, and for the quasi hyperbolic momentum algorithm in discrete-time settings.

## 1 INTRODUCTION

Most of the classical methods in convex optimization can be viewed from the perspective of a sequential approximation strategy, where a challenging optimization problem is tackled by decomposing it into a series of simpler, more manageable sub-problems that can be efficiently solved, often in closed form. Specifically, we focus on the algorithms that calculate the step direction by minimizing a surrogate function that approximates the objective in the vicinity of the current estimate. For example, the Gradient Descent algorithm (GD) determines the next estimate by minimizing an isotropic quadratic upper bound that comes from the smoothness assumption. Newton's method, on the other hand, minimizes the second-order Taylor approximation of the objective at the current estimate. Meanwhile, the Frank-Wolfe algorithm moves its estimate toward the minimizer of the first-order Taylor approximation over the problem domain.

The notable exceptions to this rule include the accelerated first-order methods with momentum. One of the earliest algorithms in this category is the Heavy-Ball (HB) method. Though it is known that HB can fail to find a solution even for certain strongly convex objectives (Lessard et al., 2016), Nesterov (1983; 2003) addressed this limitation in his renowned accelerated gradient method (NAG) by adding a correction term at the gradient step. To our knowledge, NAG is the first optimal first-order method in the sense that it achieves the worst-case iteration complexity bounds for smooth and strongly convex optimization problems up to a constant factor. While the method is known for its remarkable performance in both theory and practice, the acceleration phenomenon itself is found mysterious by many, primarily due to the absence of an easy-to-interpret sequential approximation strategy perspective for this algorithm, together with the algebraic complexity of Nesterov's original analysis, which is known as the estimate sequence technique in the literature.

Over the last few years, there has been an intense effort to understand better the essence of acceleration by analyzing the NAG algorithm through different perspectives. We can refer to (Attouch et al., 2000; Alvarez et al., 2002; Chen & Luo, 2019; Shi et al., 2019; Siegel, 2019; Laborde & Oberman, 2020; Attouch et al., 2021; Zhang et al., 2021; Chen & Luo, 2021; Ahn & Sra, 2022; Chen et al., 2022b) and the references therein, just to name a few examples. One particular camp, initiated by Alvarez et al. (2002) and revived recently by Su et al. (2016), focuses on the continuous-time analysis of acceleration, in the limits of an infinitesimal step-size.

In continuous-time analysis, the dynamics of both HB and NAG methods are often modeled by the same Polyak's damped oscillator equation. This presents a significant limitation, as these algorithms exhibit different convergence characteristics in discrete time. With the aim of incorporating the distinct behaviour of these two algorithms into continuous-time analysis, Shi et al. (2021) introduced the High-Resolution ODE framework (HR-ODE). This framework employs a separate ODE for each method, providing a more accurate model of each algorithm compared to the prior art. The key factor enabling the representation of NAG's distinct behavior appears as the inclusion of a Hessian term in the respective HR-ODE. In a follow-up work, Shi et al. (2019) discretized this HR-ODE utilizing various numerical integrators, and they demonstrated in particular that the Semi-Implicit Euler discretization (SIE) leads to an accelerated convergence rate. However, their analysis has some clear limitations. Firstly, the SIE discretization of their HR-ODE does not precisely replicate the NAG algorithm. Furthermore, while it achieves acceleration, it does not match the rate demonstrated by Nesterov (1983) for NAG for strongly convex functions. Lastly, the Lyapunov functions in their analysis lack an insightful connection to the algorithms or their respective ODEs, making it challenging to extend these results and design Lyapunov functions for HR-ODEs in general.

Our main contribution in this paper is a new general HR-ODE that successfully addresses these aforementioned limitations. More specifically, we propose the following model:

$$\begin{cases} \dot{X}_t = -m\nabla f(X_t) - n(X_t - V_t), \\ \dot{V}_t = -p\nabla f(X_t) - q(V_t - X_t). \end{cases} \qquad \text{(GM}^2\text{-ODE)}$$

Our model not only contains many of the existing ODEs in the literature for first-order accelerated methods as special cases (see Table 1), but also exhibits faster convergence rates than the prior art in (Shi et al., 2021) (see Section 3.2). At the core of our analysis is the design of a general Lyapunov function, which we crafted by leveraging Integral Quadratic Constraints (IQCs) from control theory (see Section Section 3.1). Thanks to this general Lyapunov function, we analyzed several other important ODEs and algorithms, and in particular we got faster convergence rates for the Triple Momentum Method (TMM) ODE than what was achieved in (Sun et al., 2020) (see Section 3.3). In addition, we demonstrate that various accelerated methods from the literature can be derived through the SIE discretization of our (GM$^2$-ODE) model (see Table 1). In contrast with other existing HR-ODEs, our model precisely recovers the NAG algorithm of (Nesterov, 1983) via SIE discretization (see Section 5.2). Moreover, we also improve on the convergence rate for the Quasi Hyperbolyc Momentum (QHM) algorithm (Ma & Yarats, 2019) (see Section 4). We provide a more detailed discussion on the comparison of our work with the existing results in Section 5.

**Problem template and notation.** We consider the following unconstrained optimization template:

$$\min_{x\in\mathbb{R}^n} f(x). \qquad (1)$$

Throughout, we assume that $f : \mathbb{R}^n \to \mathbb{R}$ is $\mu$-strongly convex and $L$-smooth. We denote the condition number of $f$ by $\kappa := L/\mu$. The problem has a unique solution due to strong-convexity. We denote the solution by $x^*$ and the optimal value by $f^* := f(x^*)$.

**The NAG Algorithm.** Given an initial state $x_0 = y_0 \in \mathbb{R}^n$ and a step-size parameter $s > 0$, the NAG algorithm updates the variables $x_k$ and $y_k$ iteratively as follows:

$$\begin{aligned} y_{k+1} &= x_k - s\nabla f(x_k), \\ x_k &= y_k + \alpha(y_k - y_{k-1}). \end{aligned} \qquad (2)$$

For $s = \frac{1}{L}$ and $\alpha = \frac{1-\sqrt{1/\kappa}}{1+\sqrt{1/\kappa}}$, NAG guarantees that $f(x_k) - f^* \leq \mathcal{O}\big((1 - \sqrt{1/\kappa})^k\big)$.

## 2 RELATED WORK

This section presents an overview of the related works and main approaches in continuous-time analysis focusing on accelerated methods. More detailed and technical comparisons of our approach with relevant results from the literature are provided throughout the paper as needed.

Research in this field can be traced back to (Alvarez et al., 2002), where the authors integrated a continuous Newton dynamical system with the heavy ball with friction, resulting in a combined inertial system that retains the key advantages of both approaches.

Recently, Su et al. (2016) proposed an ODE that models continuous-time dynamics of the NAG algorithm. Through this perspective, they introduced the so-called *speed restarting scheme* and further improved the convergence rate of NAG as a momentum-based method. This result has revived the interest in analyzing dissipative ODEs for accelerated methods, such as Polyak's ODE.

Building on this line of research, Wibisono et al. (2016) showed that new accelerated algorithms can be derived through discretization of these ODEs. More specifically, they used *Bregman Lagrangian* to construct a second-order accelerated ODE in non-Euclidean setting. Although the Explicit Euler discretization is unstable with this ODE, they were able to derive a stable accelerated method in the non-Euclidean setting via the so-called *rate matching discretization*. From a high-level perspective, their algorithm combines accelerated higher-order gradient methods with mirror descent steps.

More recently, Wilson et al. (2021) introduced a new variant of the Bregman Lagrangian and developed a unified analysis of several existing accelerated algorithms. Their analysis involves establishing an equivalence between the estimate sequence technique of Nesterov (1983) and a family of Lyapunov functions, applicable in both continuous and discrete time settings.

In a related vein, Siegel (2019) focused on damped Hamiltonian dynamics with a potential energy function to provide an interpretation of acceleration. More specifically, they used a forward Euler step followed by a SIE step with perturbation to derive an accelerated algorithm. They also extended the analysis to non-smooth functions, and to a setting with randomness in the discretization scheme.

Zhang et al. (2018) successfully applied Runge Kutta discretization on the initial low-resolution high-order ODE in (Wibisono et al., 2016) to achieve acceleration. Their analysis relies on a special smoothness condition. In a follow-up, Chen & Li (2020) improved their results in several directions.

Following the same line as (Su et al., 2016), Shi et al. (2021) proposed high-resolution ODEs by considering higher order terms while deriving the ODE in limit. The higher order term was then approximated and an accelerated method through the SIE discretization was found (Shi et al., 2021). In (Zhang et al., 2021), a general framework on high-resolution ODEs was proposed. They showed that both the SIE and EE integrators are capable of achieving acceleration. They also derive convergence rates for various methods like the NAG and QHM algorithms.

In (Attouch et al., 2020; 2021), asymptotic vanishing damping is combined with Hessian-driven damping which uses Hessian approximation after discretization. It is noteworthy that the latter Hessian-driven terms are from second-order information of Newton's method, while the gradient correction in (Shi et al., 2021) entirely depends on the first-order information (Chen et al., 2022a).

Lastly, Muehlebach & Jordan (2021) interpreted NAG through a dynamical system perspective, and derived the NAG method through SIE discretization of a mass-spring damping system. The major novelty of their ODE is a curvature dependent damping term, which was later considered as an essential element of acceleration under the terminology *implicit velocity* in (Chen et al., 2022a). In addition, non-smooth dynamical systems were used to provide insights on first order accelerated methods in constrained optimization (Schechtman et al., 2022; Muehlebach & Jordan, 2023).

## 3 CONTINUOUS-TIME ANALYSIS

In this section, we investigate the convergence results for (GM$^2$-ODE). Specifically, we propose a general form of Lyapunov function that shows convergence of the (GM$^2$-ODE).

### 3.1 CONSTRUCTION OF THE LYAPUNOV FUNCTION

We employ a Lyapunov function to prove the convergence of the trajectory of (GM$^2$-ODE) towards the optimal solution $x^*$. This function remains positive for all points except at the optimal solution, where it reaches zero, and it exhibits a decreasing trend along the trajectory $X(t)$ (Khalil, 2002). In the context of accelerated methods, Lyapunov functions are commonly used to study the convergence rate and stability of the underlying algorithm (Wibisono et al., 2016; Wilson et al., 2021; Shi et al., 2021; 2019; Laborde & Oberman, 2020; Su et al., 2016; Attouch et al., 2020). Recently, Sanz Serna & Zygalakis (2021) highlighted the connections between the Lyapunov functions and the optimization algorithms for the Polyak's ODE based on the results in (Fazlyab et al., 2018), in

Table 1: Various recovered ODEs and algorithms from (GM$^2$-ODE) and (13). (HB-ODE) is calculated for $\alpha = \frac{1-\sqrt{\mu s}}{1+\sqrt{\mu s}}$. For HNAG we considered $\gamma$ to be a constant.

| ODE/Algorithm | ODE (GM$^2$-ODE) | Algorithm (13) |
|---|---|---|
| (HB-ODE)/ HB | $n = q = \sqrt{\mu}$, 
 $m = 0, p = \frac{1}{\sqrt{\mu}} + \sqrt{s}$ | $n = q = \frac{1-\alpha}{\sqrt{s}(1+\alpha)}$, 
 $m = 0, p = \frac{1}{n} + \sqrt{s}$, |
| (NAG-ODE)/NAG (2) | $n = q = \sqrt{\mu}$, 
 $m = \sqrt{s}, p = \frac{1}{\sqrt{\mu}}$ | $n = q = \sqrt{\mu}$, 
 $m = \sqrt{s}, p = \frac{1}{\sqrt{\mu}}$ |
| Gradient Flow/ GD | $n = 0, m = 1$, 
 any $p, q$ | $n = 0, m = 1$ 
 any $p, q$ |
| (HR-TM))/ TM method | $m = \gamma\sqrt{s}(1 + \sqrt{Ms}), q = \xi\sqrt{M}$ 
 $n = (2-\xi)\sqrt{M}, p = \frac{(1-\xi\gamma\sqrt{Ms})(1+\sqrt{Ms})}{(2-\xi)\sqrt{M}}$ | $m = \frac{1}{\sqrt{L}}, n = \frac{2\sqrt{\mu L}}{\sqrt{L}-\sqrt{\mu}}$, 
 $q = \sqrt{\mu}, p = \frac{1}{\sqrt{\mu}}$ |
| Chen & Luo (2019)($\gamma = \mu$) 
 /HNAG | $m = \beta, n = 1$ 
 $q = 1, p = \frac{1}{\mu}$ | $n = \sqrt{\gamma}, q = \frac{\mu\sqrt{\gamma}}{\gamma + \mu\sqrt{\frac{\gamma}{L}}}$ 
 $p = \frac{\sqrt{\gamma}}{\gamma + \mu\sqrt{\frac{\gamma}{L}}}, m = \frac{1}{\sqrt{L}}$ |
| Polyak's ODE | $q = n = \sqrt{\mu}, m = 0, p = \frac{1}{\sqrt{\mu}}$ | |

Table 2: Comparison of various rates proposed in this work with prior results.

| | (Shi et al., 2021) | (Zhang et al., 2021) | (Sun et al., 2020) | HNAG(Chen & Luo, 2019) | This work |
|---|---|---|---|---|---|
| NAG-ODE | $\mathcal{O}\left(e^{-\frac{\sqrt{\mu}t}{4}}\right)$ | $\mathcal{O}\left(e^{-\frac{\sqrt{\mu}t}{2}}\right)$ | - | - | $\mathcal{O}(e^{-\sqrt{\mu}t})$ |
| NAG Algorithm | $\mathcal{O}\left((1+\frac{1}{9}\sqrt{\frac{1}{\kappa}})^{-k}\right)$ | $\mathcal{O}\left((1+\frac{1}{30}\sqrt{\frac{1}{\kappa}})^{-k}\right)$ | - | $\mathcal{O}\left((1+\frac{\sqrt{\kappa}-1}{\kappa\sqrt{\kappa}})^{-k}\right)$ | $\mathcal{O}\left((1-\sqrt{\frac{1}{\kappa}})^k\right)$ |
| HR-TM | - | - | $\mathcal{O}\left(e^{-\frac{\sqrt{M}t}{2}}\right)$ | - | $\mathcal{O}\left(e^{-\frac{2\sqrt{M}t}{3}}\right)$ |
| QHM | - | $\mathcal{O}\left((1+\frac{1}{40}\sqrt{\frac{1}{\kappa}})^{-k}\right), \kappa \geq 9$ | - | - | $\mathcal{O}\left((1-\sqrt{\frac{1}{3\kappa}})^k\right)$ |

both continuous and discrete settings. Building on these connections, we introduce our Lyapunov function in the next theorem.

**Theorem 3.1** (Continuous Convergence). *Assume $f$ is a $\mu$-strongly convex function. Further, suppose $n, m, q \geq 0$, $p > 0$, $n = q$, and $q \leq p\mu$. Then a point $(x^*, x^*) \in \mathbb{R}^n$ is globally asymptotically stable for (GM$^2$-ODE) as*

$$\varepsilon(X_t, V_t) \leq e^{-qt}\varepsilon(X_0, V_0) \quad with \quad \varepsilon(t) = f(X(t)) - f(x^*) + \frac{q}{2p}\|V - x^*\|^2. \tag{3}$$

We present the proof of this result in Appendix B.2, and we provide a brief discussion on how to deduce an exponential convergence rate from the Lyapunov function in Appendix A.3. The fastest convergence rate conditions are alleviated using a Grönwall-based result proposed in Section 3.2.

Our Lyapunov function in (3) represents a non-trivial extension of the function introduced in Theorem 4.3 of (Sanz Serna & Zygalakis, 2021), a point we will elaborate on in the following discussion.

**Remark 3.1.1** (High-Resolution Lyapunov Function Extension). *Let us rewrite (3) by using the expansion of $V$ through (GM$^2$-ODE) and obtain*

$$\varepsilon(t) = f(X(t)) - f(x^*) + \frac{q}{2p}\left\|\frac{\dot{X}}{n} + X - x^* + \frac{m}{n}\nabla f(X)\right\|^2. \tag{4}$$

*This extends Theorem 4.3 in (Sanz Serna & Zygalakis, 2021) with a high-resolution correction term. Note that setting $m = 0$ results in a low-resolution ODE, eliminating the additional gradient term and leading to the result presented in (Sanz Serna & Zygalakis, 2021).*

### 3.2 IMPROVED CONVERGENCE RATE

Next, in Theorem 3.2, we extend Theorem 3.1 for the setting $n \geq q$. To this end, we also extend the Lyapunov function discussed in Section 3.1. The proof of the theorem, which is based on Grönwall's lemma (Khalil, 2002), can be found in Appendix B.3.

**Theorem 3.2** (Continuous Convergence (Grönwall)). *For $\mu$-strongly convex $f$ and the system of ODEs as (GM$^2$-ODE) with $p, n > 0, m, q \geq 0, n \geq q, q/p \leq \mu$ we have*

$$\varepsilon(X(t), V(t)) \leq e^{-qt}\varepsilon(X(0), V(0)), \quad \varepsilon(t) = f(X) - f(x^*) + \frac{n}{2p}\|V - x^*\|^2,$$

*for the Lyapunov function $\varepsilon(t)$ and any $V(0), X(0)$. This implies that $\exists C > 0$ such that $f(X(t)) - f(x^*) \leq Ce^{-qt}$.*

Note that Theorem 3.2 does not require the smoothness of $f$. Also, with various parameter choices infinitely many ODEs can be found to achieve the same rate of convergence as (NAG-ODE). For example, for a fixed $q$, increasing $p$ would imply different ODEs, but all of them have the same convergence rate (since $q$ is fixed). We now use the improved result to prove a faster convergence rate for the TM method's high-resolution ODE in the next sub-section.

### 3.3 IMPROVED CONVERGENCE RESULT FOR TRIPLE MOMENTUM METHOD'S ODE

The TM method is known as the fastest first-order method for $f \in \mathcal{F}_{\mu,L}$ (Van Scoy et al., 2018). The state space presentation of TM method[1] is

$$\begin{cases} \epsilon_{k+1} &= (1+\beta)\epsilon_k - \beta\epsilon_{k-1} - s\nabla f(y_k), \\ y_k &= (1+\gamma)\epsilon_k - \gamma\epsilon_{k-1}, \\ x_k &= (1+\delta)\epsilon_k - \delta\epsilon_{k-1}, \end{cases} \qquad \text{(TM-Method)}$$

where the parameters are $(s, \beta, \gamma, \delta) = \left(\frac{1+\rho}{L}, \frac{\rho^2}{2-\rho}, \frac{\rho^2}{(1+\rho)(2-\rho)}, \frac{\rho^2}{1-\rho^2}\right)$, with the initial conditions $\epsilon_0, \epsilon_{-1} \in \mathbb{R}^n$, $x_k \in \mathbb{R}^n$ as the output and $\rho = 1 - \sqrt{1/\kappa}$ with $\kappa = L/\mu$. In a recent analysis, the HR-ODE of the TM method was proposed using $y_k = Y(t_k), t_k = k\sqrt{s}$ (Sun et al., 2020),

$$\ddot{Y}_t + 2\sqrt{M}\dot{Y}_t + \gamma(1 + \sqrt{Ms})\sqrt{s}\nabla^2 f(Y_t)\dot{Y}_t + (1 + \sqrt{Ms})\nabla f(Y_t) = 0, \qquad \text{(HR-TM)}$$

where $M = \left(\frac{1-\beta}{\sqrt{s}(1+\beta)}\right)^2$ is a $\kappa$ dependent constant larger than $\mu$. This ODE is different from (NAG-ODE) because they have different coefficient of Hessian term. For more information on the derivation of (HR-TM) see (Sun et al., 2020). Here, we derive an improved convergence result for the (HR-TM) through Theorem 3.2. See Appendix B.5 for the proof.

**Corollary 3.2.1** (Improved Convergence Rate on TM method's ODE). *(GM$^2$-ODE) reduces to (HR-TM) with the choice of $m = \gamma\sqrt{s}(1 + \sqrt{Ms}), q = \xi\sqrt{M}, n = (2 - \xi)\sqrt{M}$, $p = \frac{(1-\xi\gamma\sqrt{Ms})(1+\sqrt{Ms})}{(2-\xi)\sqrt{M}}$, and $\xi$ is a universal constant smaller than $2/3$. Furthermore, invoking Theorem 3.2 with parameters stated above and $C_{TM} > 0$, leads to*

$$f(Y(t)) - f(x^*) \leq C_{TM}e^{-\xi\sqrt{M}t} \qquad t \geq 0. \tag{5}$$

**Remark 3.2.1** (Comparison with Theorem 4.1 in (Sun et al., 2020)). *The previous rate for (HR-TM) was reported inSun et al. (2020). This rate was $f(Y(t)) - f(x^*) \leq C_{TM}^1 e^{-p_{TM}^* Mt}$ where $C_{TM}^1 > 0$ is a constant and $p_{TM}^* \leq \sqrt{M}/2$. Here, we improved this rate to $f(Y(t)) - f(x^*) \leq C_{TM}e^{-\frac{2}{3}\sqrt{M}t}$ (see Figure 1).*

## 4 DISCRETE-TIME ANALYSIS

### 4.1 GENERAL DISCRETE CONVERGENCE RESULTS

We now build on our earlier results and propose a new general accelerated algorithm. The convergence rate is calculated through a discrete Lyapunov analysis. The choice of the Lyapunov function is based on the continuous counterpart in Theorem 3.2.

Applying the SIE discretization to (GM$^2$-ODE) gives

$$\begin{cases} x_{k+1} - x_k = &-m\sqrt{s}\nabla f(x_k) - n\sqrt{s}(x_{k+1} - v_k), \\ v_{k+1} - v_k = &-p\sqrt{s}\nabla f(x_{k+1}) - q\sqrt{s}(v_k - x_{k+1}), \end{cases} \tag{6}$$

---

[1]See (d'Aspremont et al., 2021) for another representation of the TM method.

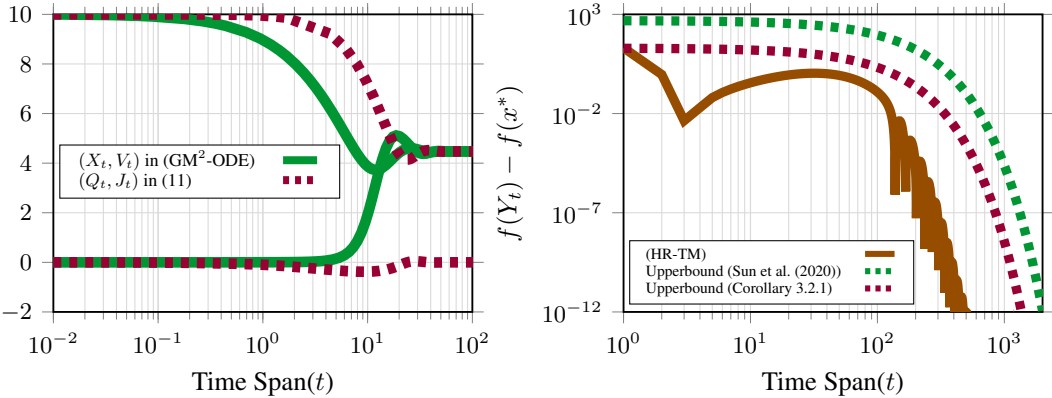

Figure 1: Continuous-time simulation using $f(x) = 4(L - \mu)\log(1 + e^{-x}) + \frac{\mu}{2}x^2$, (left) comparison of trajectories $(X_t, V_t)$ from (GM$^2$-ODE) and $(Q_t, J_t)$ from (11). The simulation is done for $L = 1, \mu = 10^{-2}$ and $X(0) = Q(0) = 10, V(0) = J(0) = 0$, (right) trajectory of (HR-TM) ODE for $L = 10, \mu = 10^{-3}$, random $Y(0), \dot{Y}(0) = 0$ and its corresponding upper bounds.

Theorem 4.1 shows the convergence rate of (6). Specifically, we will use Lyapunov function

$$\varepsilon(k) = f(x_k) - f(x^*) + \frac{B}{2}\|v_k - x^*\|_2^2 - \frac{Bp^2s}{2}\|\nabla f(x_k)\|^2, \tag{7}$$

to show convergence. The proposed Lyapunov function converges to its continuous counterpart as step-size $s \to 0$. The choice of existing terms in the Lyapunov function are mainly the same as the continuous Lyapunov function except the last term. Note that adding smoothness assumption ensures the positivity of the Lyapunov function. Similar technical terms are seen in previous works (see (Shi et al., 2019; d'Aspremont et al., 2021; Zhang et al., 2021)).

Unlike (Fazlyab et al., 2018), the last term in the Lyapunov function (7) is not always positive. Thus, we cannot directly deduce a convergence rate for $f(x_k) - f(x^*)$. Nevertheless, we can establish a convergence rate for $\|v_k - x^*\|^2$ which can later be used to show the convergence rate of $f(x_k) - f(x^*)$. We now have the following theorem and the proof is provided in Appendix B.4.

**Theorem 4.1.** *Consider update rule (6) to solve (1), for a $\mu$-strongly convex $L$-smooth function $f$. If we have $q/p \leq \mu$, $0 \leq nps \leq m\sqrt{s} \leq 1/L$, $0 \leq q\sqrt{s} < 1$, $p > 0$, and $n = q$, then the Lyapunov function (7) with $B = \frac{n}{p}$ decreases as $\varepsilon(k) \leq (1 - q\sqrt{s})^k \varepsilon(0)$. Furthermore, we have*

$$f(x_k) - f(x^*) \leq C''_{GM}(1 - q\sqrt{s})^k,$$

*for constant $C''_{GM} > 0$ and any $x_0, v_0 = x_0 - \frac{m}{n}\nabla f(x_0)$.*

It is noteworthy that (Zhang et al., 2021) proposed a convergence rate for the QHM method (Ma & Yarats, 2019) as a special case of their convergence results. We achieve a faster convergence rate through Theorem 4.1. This result is summarized in the following corollary and its proof is provided in Appendix B.6.

**Corollary 4.1.1.** *Consider $\mu$-strongly convex $L$-smooth function $f$. Let $m = (1 - a)\sqrt{s}$, $p = \frac{a}{q} + \sqrt{s}$, $n = q = \sqrt{a\mu}$, $0 < a \leq \frac{1}{4}$, $q\sqrt{s} \leq \frac{1}{2}$, $s \leq \frac{4}{3L}$. Then, (6) reduces to*

$$\begin{cases} x_{k+1} - x_k = -s(1-a)\nabla f(x_k) - sa(g_{k+1}), \\ g_{k+1} = bg_k + \nabla f(x_k), \end{cases} \tag{QHM}$$

*with parameters $a$ and $b = \frac{1 - q\sqrt{s}}{1 + q\sqrt{s}}$. Also, the convergence rate of (QHM) is $f(x_k) - f(x^*) \leq C_{QHM}(1 - \sqrt{a\mu s})^k$ for a positive constant $C_{QHM}$.*

The result proposed in Corollary 4.1.1 shows faster rate of convergence than what was previously proved in (Zhang et al., 2021) (which was $(1 + \sqrt{1/\kappa}/40)^{-k}$). The improved rate is depicted through numerical experiments in Figure 2. Interestingly, existence of $a$ in the choice of $m$ and $p$, leads to the possibility of increasing the step-size up to $2/\sqrt{3L}$ which is higher than the step-size for the NAG algorithm.

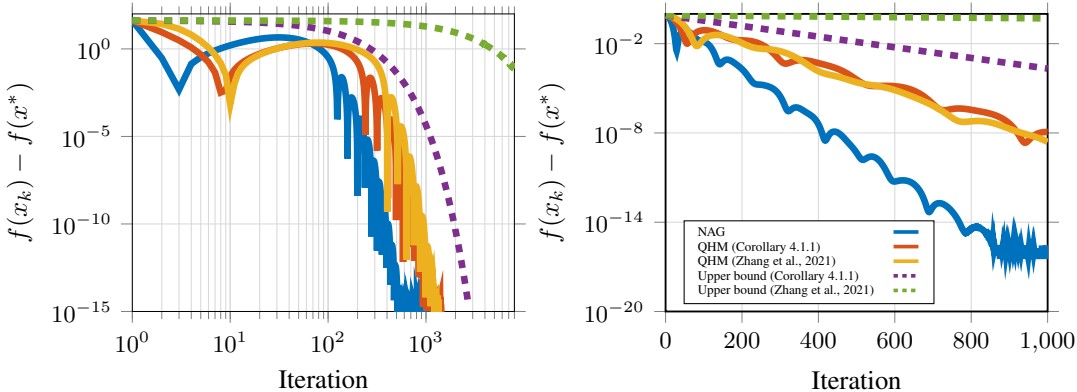

Figure 2: Discrete-time simulation; comparison of QHM method performance under various settings with existing upper bounds for (a) $f(x) = 4(L - \mu)\log(1 + e^{-x}) + \frac{\mu}{2}x^2$, $L = 10, \mu = 10^{-3}$ and (b) 10-dimensional regularized binary classification problem with logistic loss for random data and labels of length 1000. The regularization parameter was $\mu = 10^{-3}$. In all experiments, parameters are chosen so that the best possible theoretical convergence rate is achieved.

## 5 DISCUSSION AND COMPARISON

We now discuss the properties of various HR-ODEs, their discretizations, and their relation to (GM²-ODE). Shi et al. (2021) found the HB and the NAG method's high-resolution ODEs as

$$\ddot{X}_t + 2\sqrt{\mu}\dot{X}_t + (1 + \sqrt{\mu s})\nabla f(X_t) = 0, \tag{HB-ODE}$$

$$\ddot{X}_t + 2\sqrt{\mu}\dot{X}_t + \sqrt{s}\nabla^2 f(X_t)\dot{X}_t + (1 + \sqrt{\mu s})\nabla f(X_t) = 0. \tag{NAG-ODE}$$

Then, Shi et al. (2019) discretized these ODEs using various discretizers and showed their convergence rates. Unfortunately, their convergence rates for the (NAG-ODE) and the NAG algorithm do not achieve the same convergence guarantees, considering the constant factors, as the original method. In addition, after discretizing (NAG-ODE) using the SIE integrator, the resulting method mildly differs from the NAG algorithm (Shi et al., 2021). In a similar vein, Zhang et al. (2021) showed that proper choice of the high-resolution ODE and the discretizer results in an accelerated method[2]. This was done through the general ODE

$$\begin{cases} \dot{U}_t &= -m'\nabla f(U_t) - n'W_t, \\ \dot{W}_t &= \nabla f(U_t) - q'W_t, \end{cases} \tag{GM-ODE}$$

with $m', n', q' \geq 0$, $U_t = U(t)$, and $W_t = W(t)$. Though the success they had in showing the possibility of acceleration through different discretizers, their analysis cannot recover the same convergence rate as the NAG's.

### 5.1 CONVERGENCE RATES

**The NAG Algorithm** Taking $n = \sqrt{\mu}, q = \sqrt{\mu}, p = \frac{1}{\sqrt{\mu}}, m = \sqrt{s}$ reduces (6) to the NAG algorithm. In this case, Theorem 4.1 gives the following convergence rate $f(x_k) - f(x^*) \leq C''_{GM}(1 - \sqrt{1/\kappa})^k$. This shouldn't be a surprise since the rate matches the well-known rate for the NAG method. However, this convergence rate is faster than the ones proven for the numerical discretizations in (Shi et al., 2021) $(f(x_k) - f(x^*) \leq C'(1 + \sqrt{1/\kappa}/9)^{-k})$ and (Zhang et al., 2021) $(f(x_k) - f(x^*) \leq C''(1 + \sqrt{1/\kappa}/30)^{-k})$. We believe this is for two reasons. First, note that the phase space representation used by Shi et al. (2021) cannot exactly recover the NAG algorithm after discretization. This is further discussed in the next sub-section. Second, they have used a sub-optimal Lyapunov function in their analyses.

---

[2]Similar result holds for (GM²-ODE). The acceleration through Explicit Euler (EE) discretization of (GM²-ODE) is investigated in Appendix A.4.

The Hessian-driven Nesterov Accelerated Gradient algorithm (HNAG, see equation (42) in (Chen & Luo, 2019)) achieves a convergence rate of $\mathcal{O}\big((1 + \sqrt{\min\{\gamma, \mu\}/L})^{-k}\big)$. For comparison, we consider the specific scenario where HNAG recovers NAG. In this case, $\gamma = \mu(1 - \alpha)$ for some particular $\alpha \geq \sqrt{1/\kappa}$, and the rate becomes $\mathcal{O}\big((1 + \sqrt{(1-\alpha)/\kappa})^{-k}\big)$. Our method in (6) can recover HNAG (see Table 1), in this case, we will have $q = n = \sqrt{\gamma} < \sqrt{\mu}$, leading to a convergence rate of $\mathcal{O}\big((1 - \sqrt{(1-\alpha)/\kappa})^k\big)$. This is clearly an improvement since

$$1 - \sqrt{(1-\alpha)/\kappa} \leq \frac{1 - \sqrt{(1-\alpha)/\kappa}}{1 - (1-\alpha)/\kappa} = \frac{1}{1 + \sqrt{(1-\alpha)/\kappa}}. \tag{8}$$

In addition, we can also recover NAG directly, eliminating the need for the specific parameter choices required for recovery through HNAG. In this case, we further improve the rate to $\mathcal{O}\big((1 - \sqrt{1/\kappa})^k\big)$, which matches the original rates shown in (Nesterov, 1983).

**Best Possible Rate**   Theorem 4.1 suggests various convergence rates for different choices of parameters. One may wonder if a better convergence rate than the NAG's is possible or not. Take $s = ((1-\rho)/q)^2$ so that $\rho = 1 - q\sqrt{s}$. Plugging this in the conditions of Theorem 4.1 yields $\rho \geq \max\{1 - m/p, 1 - q/(mL)\}$. These two values are oppositely controlled by $m$. Thus, the equality of these will determine the solution for $m = \sqrt{(qp)/L}$. Replacing $m$ in the bounds gives $\rho \geq 1 - \sqrt{q/(pL)}$. The r.h.s cannot be smaller than $1 - \sqrt{\mu/L}$ since $q/p \leq \mu$.

**GM$^2$-ODE vs GM-ODE**   It is important to understand the connection between (GM$^2$-ODE) and (GM-ODE). Proposition 5.1 shows the correspondence between these two frameworks. The proof is based on comparing the one-line representations (14) and the one-line representation of (GM$^2$-ODE)

$$\ddot{X}_t + ((n+q) + m\nabla^2 f(X_t))\dot{X}_t + (np + mq)\nabla f(X_t) = 0. \tag{9}$$

**Proposition 5.1.** *For a $\mu$ strongly-convex function $f(x)$, consider (GM-ODE) with parameters $m', n', q'$ and (GM$^2$-ODE) with parameter $m, n, p, q$. The problem of minimizing $f$ on the trajectory of (GM$^2$-ODE) is equivalent to the problem of minimizing $f$ on (GM-ODE) if*

$$m' = m, \quad n' = n(p - m), \quad q' = n + q, \quad p > m. \tag{10}$$

Next, we recall the following convergence result from (Zhang et al., 2021).

**Remark 5.1.1** (Theorem 1 (Zhang et al., 2021))**.** *For $f \in \mathcal{F}_{\mu,L}$ and $m', n', q' \geq 0$, a point $(x^*, 0) \in \mathbb{R}^{2d}$ is globally asymptotically stable for (GM-ODE) as $\varepsilon'(U_t, W_t) \leq e^{-\gamma_1 t}\varepsilon'(U_0, W_0)$, with $\varepsilon'(U_t, W_t) = (q'm' + n')(f(U_t) - f(x^*)) + \frac{1}{4}\|q'(U_t - x^*) - n'W_t\|^2 + \frac{n'(q'm'+n')}{4}\|W_t\|^2$, where $\gamma_1 := \min\left(\frac{\mu(n'+q'm')}{2q'}, \frac{q'}{2}\right)$.*

In Remark 5.1.1, $\varepsilon'(U_t, W_t)$ is their proposed continuous Lyapunov function for (GM-ODE). Using Proposition 5.1, the choice of coefficients $n' = 1 - \sqrt{\mu s}, m' = \sqrt{s}, q' = 2\sqrt{\mu}$ in (GM-ODE) recovers the (NAG-ODE). An application of Remark 5.1.1 then gives

$$f(X_t) - f(x^*) \leq C'_{GM} e^{-\gamma_1 t} \leq C'_{GM} e^{-\frac{\sqrt{\mu}}{2}t},$$

with $\gamma_1 = \min\left(\frac{\sqrt{\mu}(1 + \sqrt{\mu s})}{4}, \sqrt{\mu}\right)$, $\sqrt{\mu s} \leq 1$, and $C'_{GM} > 0$. Note that high-resolution ODEs in the limit of $s \to 0$ reduce to the Polyak's ODE. We know that the convergence rate for the Polyak's ODE is of $\mathcal{O}(e^{-\sqrt{\mu}t})$ (Wilson et al., 2021). Hence, one would expect the high-resolution ODEs to have a similar convergence rate when $s \to 0$. On the other hand, (GM$^2$-ODE) can successfully recover the (NAG-ODE) by setting $n = q = \sqrt{\mu}, p = 1/\sqrt{\mu}, m = \sqrt{s}$. Using these parameters in Theorem 3.1, results in the convergence rate $f(x(t)) - f(x^*) \leq Ce^{-\sqrt{\mu}t}$, for $C > 0$ which is twice the rate $C'_{GM} e^{-\sqrt{\mu}/2t}$ found by (Zhang et al., 2021).

## 5.2   RECOVERY OF ODEs AND ALGORITHMS

Reformulating (GM$^2$-ODE) gives

$$\begin{cases} \dot{Q}_t = J_t, \\ \dot{J}_t = -((n+q) + m\nabla^2 f(Q_t))J_t - (np + mq)\nabla f(Q_t), \end{cases} \tag{11}$$

in which $Q_t := Q(t)$ and $J_t := J(t)$ is known as the momentum term in the literature (Wibisono et al., 2016; Shi et al., 2021). This is the same phase space representation used by Shi et al. (2019) to study various discretizations. In the limit $t \to \infty$, $Q_t$ and $J_t$ converge to $x^*$ and $0$ respectively. In contrast, $V_t$ in (GM$^2$-ODE) is a coupled variable with $X_t$ which converges to $x^*$. This is shown in Figure 1. Due to the presence of a Hessian term, naively discretizing (NAG-ODE) will lead to a second-order algorithm. This issue can be resolved in two ways. Either approximating the Hessian as $\sqrt{s}\nabla^2 f(X_t)\dot{X}_t \approx \nabla f(x_{k+1}) - \nabla f(x_k)$, or rewriting (NAG-ODE) in a Hessian-free manner. The former was done by (Shi et al., 2021). Indeed, the SIE discretization of (11) using a Hessian approximation reads

$$\begin{cases} q_{k+1} - q_k = j_k\sqrt{s}, \\ j_{k+1} - j_k = -(n+q)\sqrt{s}j_{k+1} - \sqrt{s}(np+mq)\nabla f(q_{k+1}) - m(\nabla f(q_{k+1}) - \nabla f(q_k)). \end{cases} \quad (12)$$

Now, when $n = \sqrt{\mu}, q = \sqrt{\mu}, p = 1/\sqrt{\mu}, m = \sqrt{s}$, the numerical scheme (12) coincides with the one studied in (Shi et al., 2021). It is worth pointing out that this scheme is not the NAG method due to the choice of the coefficients in (12). In particular, if one now chooses $n = \frac{\sqrt{\mu}}{1-\sqrt{\mu s}}, q = \frac{\sqrt{\mu}}{1-\sqrt{\mu s}}, p = \frac{1}{\sqrt{\mu}}, m = \sqrt{s}$, (12) would coincide with the NAG method. The second solution is to avoid the Hessian term by using (GM$^2$-ODE) instead of (11) for discretization. Applying the SIE discretization to (GM$^2$-ODE) results in

$$\begin{cases} x_{k+1} - x_k = & -m\sqrt{s}\nabla f(x_k) - n\sqrt{s}(x_{k+1} - v_k), \\ v_{k+1} - v_k = & -p\sqrt{s}\nabla f(x_{k+1}) - q\sqrt{s}(v_k - x_{k+1}), \end{cases} \quad (13)$$

which reduces to the NAG algorithm by setting $n = \sqrt{\mu}, q = \sqrt{\mu}, p = 1/\sqrt{\mu}, m = \sqrt{s}$. This shows that the formulation (13) exactly recovers the NAG algorithm with the same coefficients that (GM$^2$-ODE) needs to recover (NAG-ODE). This paves the way for a more consistent Lyapunov analysis and essentially, better convergence rates in both continuous and discrete setups.

**GM-ODE** Zhang et al. (2021) proposed (GM-ODE) as a general framework for first-order accelerated algorithms in smooth strongly convex regime. Indeed, one can write the one line ODE of (GM-ODE)

$$\ddot{U}_t + (q' + m'\nabla^2 f(U_t))\dot{U}_t + (n' + m'q')\nabla f(U_t) = 0. \quad (14)$$

Zhang et al. (2021) sets $m' = \sqrt{s}, q' = 2\sqrt{\mu}, n' = 1$ to recover the NAG's high-resolution ODE. However, these coefficients will lead to

$$\ddot{U}_t + (2\sqrt{\mu} + \sqrt{s}\nabla^2 f(U_t))\dot{U}_t + (1 + 2\sqrt{\mu s})\nabla f(U_t) = 0, \quad (15)$$

and not exactly (NAG-ODE) as they differ in the coefficient of $\nabla f(U_t)$. Also, the SIE discretization of (15) is not the same as the NAG method as they set $m' = \sqrt{s}, q' = 2\sqrt{\mu}, n' = 1 - 2\sqrt{\mu s}$ in (14) to recover the NAG algorithm. This results in inconsistency of their model in recovery of ODEs and algorithms. This is not the case for (GM$^2$-ODE) as shown in Table 1.

## 6 CONCLUSION AND FUTURE DIRECTIONS

In this work, we focused on the continuous-time analysis for the class of accelerated gradient methods for unconstrained minimization of a smooth and strongly convex objective function. More specifically, we proposed (GM$^2$-ODE) as a new general HR-ODE. We demonstrated that the SIE discretization of our HR-ODE exactly reproduces Nesterov's algorithm. Additionally, our HR-ODE allows for deriving several other accelerated methods from the literature by adjusting coefficients and applying SIE discretization. We investigated the long-time properties of (GM$^2$-ODE) by using a Lyapunov function based on Integral Quadratic Constraints from robust control theory. This allowed us to obtain improved convergence rates not only for the HR-ODE corresponding to NAG but also for the one corresponding to the TM method. Furthermore, extending this analysis to discrete-time allowed us to achieve better convergence rates for the QHM method.

There are certain limitations to our study which form a basis for the future directions. So far, we focused only on the strongly-convex problems. Relaxing this assumption to develop a general framework for unconstrained convex optimization would be a valuable extension of our work. Additionally, our analysis has been limited to deterministic methods. However, in practical applications, we often need to deal with noise and stochasticity. Therefore, we plan to explore the adaptability of our framework in such scenarios in future research.

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
