# OpenReview forum: "Revisiting High-Resolution ODEs for Faster Convergence Rates"
_ICLR.cc/2024/Conference — Submitted to ICLR 2024_

### Official Review · Reviewer_ut71 · 2023-10-31

**Soundness:** 3 good
**Presentation:** 2 fair
**Contribution:** 2 fair
**Rating:** 3
**Confidence:** 4

**Summary:**

The paper proposes a general high-resolution ordinary differential equations (ODE) model to investigate the dynamics of various momentum-based optimization methods. The high-resolution ODE proposed unifies many different ODE models and leads to improvement in the convergence guarantee of several existing algorithms, such as triple momentum method in continuous setting and quasi hyperbolic momentum algorithm in discrete setting.

**Strengths:**

This work provides a high-resolution ODE framework ($\text{GM}^2$-ODE) that unifies and extends different ODEs in literature. The theoretical analysis is solid and provides some improvement over existing results of accelerated methods. The presentation of the paper is also clear to me.

**Weaknesses:**

- This work follows the line of research on understanding accelerated methods via (high-resolution) ODE. Given the vast literature on this topic, I am afraid the contribution of this work is not significant enough. Although generalization and unification are developed, the results derived here are expected and the techniques are quite standard.
- The theoretical improvements are kind of minor to me, for example, improving the constant from $1/2$ to $2/3$. The theoretical understanding based on this new high-resolution ODE does not provide any new insight on acceleration. Neither does it lead to any novel algorithms with more attractive practical performance.

**Questions:**

- As I mentioned in Weakness, I am afraid the contribution of this work is significant enough given many existing works on the same topic using almost the same analysis techniques. Could the authors justify this point? Is there any particular novelty in technical and algorithmic developments I'm missing?
- The work is focused on the theoretical analysis of existing momentum-based algorithms. I'm wondering if the understanding can help develop some new approaches leading to stronger practical performance? For example, does the best possible rate improve empirical performances in practice?
- In Figure 2, it is observed that NAG is the algorithm with fastest convergence against QHM. I'm curious if the viewpoint of the ODE developed here can provide some explanation to this phenomenon.

---

> ### Author Response · Authors · 2023-11-20
>
> Thank you for your time and comments. Below, we try to address your concerns:
>
> # Weaknesses
>
> - **There is a vast literature**: The *vast literature* on this subject justifies the relevance of our work. We believe that our contribution is both significant and timely, as it unifies various approaches within this extensive body of literature, explaining the connections between them, all while achieving improved convergence rates.
>
> - **Rates are expected**: We respectfully disagree. Please observe that proving these rates through a continuous-time analysis is a non-trivial task, and it was not clear whether it was possible. It is important to note that the rates presented in the earlier works [1,2] are known to be suboptimal in discrete and continuous-time settings, whereas in many cases we achieve the optimal rates. To further highlight this, we have enhanced the discussion about comparisons with prior work in Section 5 with Table 2 in our revision to compare our rates with existing results.
>
> - **Improvements are in the constants**: We agree that our rates are linear, just as in the prior work. However, please note that our improvements are in the *exponents* and not the *constants*. This forms a significant enhancement in comparison with the prior work.
> For example, Theorem 4.1 proves a convergence rate of order $C_{GM}(1-\sqrt{ 1/\kappa})^k)$, and the best previous result in this setting was $5L||x_0-x^*||^2_2(1+\sqrt{ 1/\kappa}/9)^{-k}$. Suppose $L=10,\mu=1,\kappa=10$ and $k=50$, then $C_{GM}(1-\sqrt{ 1/\kappa})^k \approx 2.35\times 10^{-7}\times ||x_0-x^*||_2^2$ while $5L||x_0-x^*||_2^2((1+\sqrt{ 1/\kappa}/9)^{-k}) \approx 8.8939||x_0-x^*||_2^2$.
> Similarly, for QHM, we prove a rate of order $O((1-\sqrt{\frac{1}{3\kappa}})^k)$ while the previous best rate was $O((1+\frac{1}{40\sqrt{\kappa}})^{-k})$. If $\kappa=10$ and $k=50$, then $(1-\sqrt{\frac{1}{3\kappa}})^k \approx 4.19 \cdot 10^{-5}$ while $(1+\frac{1}{40\sqrt{\kappa}})^{-k} \approx 0.6745$.
> Finally, we would like to draw your attention to the ability of (GM2-ODE) to recover other high-resolution ODEs.
> - **Leading to other ODEs**
> As shown in Table 1, (GM2-ODE) and its SIE discretization recover the TM method and its ODE through different choices of coefficients. In addition, the current existing rates on (HR_TM) in the paper are not comparable to the algorithm's convergence rate $(\mathcal O\left((1-\sqrt{\tfrac{\mu}{L}}\right)^{2k}$ which should correspond to $\mathcal{O}\left(e^{-2\sqrt{\mu}t}\right)$ in continuous time). This observation suggests a new high-resolution ODE for the TM method. Setting the same coefficients that recover the TM method in (GM2-ODE) reads
>     $$\ddot{X}_t+\sqrt{\mu}\left(\frac{3-\sqrt{\tfrac{\mu}{L}}}{1-\sqrt{\tfrac{\mu}{L}}}\right)\dot X_t +\frac{1}{\sqrt{L}}\nabla^2 f(X_t)\dot X_t+\left(\frac{2}{1-\sqrt{\tfrac{\mu}{L}}}+\sqrt{\frac{\mu}{L}}\right)\nabla f(X_t)=0.$$
> Surprisingly, if the step-size $\tfrac{1}{\sqrt{L}}\rightarrow 0$ the recent ODE reduces to$$ \ddot{X}_t+3\sqrt{\mu}\dot X_t+2\nabla f(X_t)=0$$
> which is the low-resolution ODE corresponding to the TM method [3]. We will include this discussion in the revision of our paper. .

---

> > ### Author Response · Authors · 2023-11-20
> >
> > # Answers to the questions
> > 1. We would like to emphasize that our unified model (GM2-ODE) has two attractive features: \
> > **A)** Not only does it unify the ODEs in continuous time, but also existing algorithms through SIE discretization. Surprisingly, all these algorithms are found to be well-known methods (NAG,HB,TMM, etc) and they are all recovered through the same routine. \
> > **B)** The corresponding *unifying* Lyapunov function leads to better convergence rates than the previous results obtained by continuous time analysis. To further emphasize this, we added Table 2 to compare our rates with existing results, which shows precisely the improvements. \
> > It is true that we use well-established techniques like Lyapunov analysis and Grönwall lemma. Nevertheless, we leverage these tools to achieve a family of Lyapunov functions whose form is not seen before in the literature. Crucially, our analysis is NOT a trivial extension of another work, the form of the unified Lyapunov function requires different steps in the analysis. For example, we use a limit analysis after (55) in the appendix which is different from all of the prior works. This is just one example of the many analytical and algebraic complexities that arise from the generality of the Lyapunov function we addressed in our analysis.
> > 2. Yes, we have actually demonstrated that the best possible rate setting works well in practice in our numerical experiments. Please note that the best theoretical rate coefficients lead to NAG. In Figure 2, we show that NAG outperforms other methods like QHM on a binary classification task also in practice. We will clarify this.
> > We managed to recover a large number of existing algorithms through our analysis. Although we did not design a new algorithm, we believe that our analysis has the potential to lead to the discovery of new methods as well.
> > 3. Yes! The best possible rate through Corollary 4.1.1 occurs when $a=1/4$ and $s=3/4L$. Substituting these values into the residual rate gives $f(x_{k})-f(x^*)\leq C_{QHM}(1-\sqrt{\mu/3L})^k$, which is slower than the NAG rate $f(x_{k})-f(x^*)\leq C_{QHM}(1-\sqrt{\mu/L})^k$.
> > We will clarify this in the paper.
> >
> >
> > # References
> >
> > [1] Shi, Bin, et al. "Understanding the acceleration phenomenon via high-resolution differential equations." Mathematical Programming (2021): 1-70.
> >
> > [2] Shi, Bin, et al. "Acceleration via symplectic discretization of high-resolution differential equations." Advances in Neural Information Processing Systems 32 (2019).
> >
> > [3] Kim, Jungbin, and Insoon Yang. "Convergence analysis of ODE models for accelerated first-order methods via positive semidefinite kernels." Thirty-seventh Conference on Neural Information Processing Systems. 2023.
> >
> > [4] Ma, Jerry, and Denis Yarats. "Quasi-hyperbolic momentum and Adam for deep learning." arXiv preprint arXiv:1810.06801 (2018).
> >
> > [5] Maskan, Hoomaan, Konstantinos C. Zygalakis, and Alp Yurtsever. "A Variational Perspective on High-Resolution ODEs." Thirty-seventh Conference on Neural Information Processing Systems. 2023.

---

### Official Review · Reviewer_fWEa · 2023-10-31

**Soundness:** 3 good
**Presentation:** 3 good
**Contribution:** 3 good
**Rating:** 8
**Confidence:** 2

**Summary:**

This paper proposes a high resolution ODE (GM2-ODE) for analyzing accelerated gradient descent (AGD) for convex optimization. A Lyapunov function based on integral quadratic control is derived to analyze the stability and convergence rate of GM2-ODE. Semi-Implicit Euler discretization (SIE) of the ODE recovers the accelerated gradient algorithms and the known optimal convergence rates. Many previous ODEs for accelerated gradient descent can be formulated into the proposed GM2-ODE form and the convergence rates
 can be obtained using their results.

**Strengths:**

The paper is well written. The presentation of their results is clear and sound.

siginificance:
The proposed GM2-ODE enjoys intuitive form and design of Lyapunov function. The discrete time convergence rates based of the continuous time Lyapunov function recovers the optimal convergence rate of accelerated gradient method. The analysis framework applies to many previous ODEs for accelerated gradient methods and recover (even enhance) the discrete-time convergence rates.

**Weaknesses:**

NA

**Questions:**

NA

---

> ### Author Response · Authors · 2023-11-20
>
> Thank you for your positive evaluation and comments on our paper.

---

### Official Review · Reviewer_oAou · 2023-10-31

**Soundness:** 2 fair
**Presentation:** 2 fair
**Contribution:** 2 fair
**Rating:** 1
**Confidence:** 5

**Summary:**

The authors propose a unified framework to analyze the high-resolution ODEs.

**Strengths:**

No

**Weaknesses:**

The high-resolution ODE was originally proposed to find the mechanism behind Nesterov's acceleration.  I have never found any new in this paper.

**Questions:**

Could you express your motion to do this paper? Could you show where are the new parts beyond the current research?

---

### Official Review · Reviewer_6yy2 · 2023-11-04

**Soundness:** 3 good
**Presentation:** 3 good
**Contribution:** 1 poor
**Rating:** 3
**Confidence:** 4

**Summary:**

The approach of this paper is (1) to provide a unifying high-resolution ordinary differential equations (HR-ODEs) to several ones in the literature for momentum-based methods for minimization, and then (ii) to use a tool from control theory called integral quadratic constraints (IQC) to derive a Lyapunov function used for convergence analyses.

For strongly convex and smooth functions, it:
- achieves a faster convergence rate for the triple momentum method,
- achieves a faster rate for the Quasi Hyperbolic Momentum method (and for a larger step size range).

**Strengths:**

- The overall idea of unifying ODE for the momentum-based HR-ODEs is interesting.
- The paper improves some of the convergence rates in the literature for accelerated methods in the strongly convex regime.

**Weaknesses:**

# Novelty \& incremental results

The main concern is the incremental contributions.
- The techniques used are well known (Lyapunov analysis, Grönwall lemma, etc), e.g. Thm. 3.2. is an instance of Gronwall lemma.
- The improved factor for NAG is only 2. The triple momentum method is not widely used; there has been less focus on improving its rate.
- *GM2-ODE.*  In terms of structuring, it is surprising that the proposed GM2-ODE is stated in the introduction. Moreover, this ODE is not derived but rather considers a set of some HR-ODEs that exist and aims to unify them in the sense that these can be seen as instances of the GM2-ODE. This is fairly straightforward to do given several ODEs as the terms that appear have already known interpretations; there's no discussion or further development if this ODE is general enough to lead to other useful methods. Also, it is very similar to the existing ODE in Zhang et al. (2021), see eq. GM-ODE in the main part. Considering all, this is a fairly limited contribution stated as central/main.
- The only considered setting is (deterministic) smooth, strong convexity.

Although these contributions are interesting, they are not developed sufficiently for acceptance.

# Writing

The paper reads well, and I enjoyed reading it. However, content-wise, it is not on point regarding the actual focus of this paper / exact contributions / motivation for these contributions, etc. It often focuses on general optimization comments that are enjoyable to read but perhaps more suitable for a textbook, etc., and due to that, it is not concise in bringing the reader to the actual contributions and their motivation. For example:
- Abstract. A large part focuses on general comments about HR-ODEs or the Lyapunov function, which is an intermediate step of proving convergence that many methods can be seen as a discretization of the ODE, which is often the case.
   - Importantly, it leaves very unclearly what precisely the "improved convergence guarantees compared to prior art" are -- it would be helpful to state precisely if the constants or the order is improved and by what factor; what is the precise advantage of this unifying HR-ODE (is it more interpretable, etc), etc.
   - It does not even mention the setting, e.g., that the results are for strongly convex functions
   - That discrete methods can be viewed as discretizations of ODEs is well known. If keeping this sentence, it is worth mentioning the type of discretization.
- Introduction. The first two paragraphs that refer to (discrete) optimization methods generally are very enjoyable to read. Still, the motivation for using continuous-time analyses is rushed, which is more relevant to this paper. The paper would benefit from reconsidering the content vs. the page limit and prioritizing better.



# Missing smoothness assumption in Thm 3.1 and unclear notations

Thm. 3.1. states that $f$ is strongly convex, but the proof relies on Thm. 6.4 in (Fazlyab et al., 2018), which uses the assumption that $f$ is also $L$ smooth. This assumption should be stated.
The proof in App. B.2. also mentions $\sigma$, which is not defined in the paper.
The curly F notation, used in the main part and Thm. A.1 was not introduced.

# Other

- The constant $C_{QHM}$ that appears in Cor. 4.1.1. is not defined

# minor comments

- missing full stop in eq. (1)
- typo: instable
- sec. 3 title should be continuous-time analysis

**Questions:**

1. The abstract highlights that different methods can be seen as discretizations of the GM2-ODE. There are many discretization methods, and many works highlight that discrete methods can be obtained of a general ODE under some discretization [1]. Could you elaborate on why your work concentrates on crafting ODE that yields the methods through the SIE discretization versus the others and why it is more validating the derived ODE versus the other? Or is it providing more consistency with the discrete analysis? I believe this is central to be discussed since it is highlighted.
2. On Page 7, with "the phase space representation [..] cannot exactly recover the NAG algorithm after discretization", which discretization do you assume here?
3. In the [2] follow-up work of Shi et al., 2021 (on which this work builds the idea of HR-ODEs), a more "consistent" way of deriving the HR-ODEs is proposed. App. A.3. of [2] points out that such derivation is more consistent because the Taylor expansion is done on all applicable terms instead of some. Does your HR-ODE unify the HR-ODEs of the NAG and HB methods derived that way?
4. Do you know if this HR-ODE will lead to better convergence rates on other setups, e.g., convex? In other words, is the benefit of modifying the ODE of Zhang et al. 2021 specific to the strongly convex setup?
5. Is the upper bound dictated by Thm 4.1. matching the known one for this setting for the constants?

-----
[1] *On dissipative symplectic integration with applications to gradient-based optimization*, França, Jordan, and Vidal, 2021.

[2] *Last-Iterate Convergence of Saddle-Point Optimizers via High-Resolution Differential Equations*, Chavdarova, Jordan, and Zampetakis, 2023.

---

> ### Author Response · Authors · 2023-11-20
>
> Thank you for your time, careful reading, and valuable comments. Our responses to the specific concerns and comments are below:
>
> # Novelty and Significance
> - **Use of IQCs**: It is true that we leveraged IQCs, but please note we use it only in one of the three Lyapunov functions we propose in this paper. We introduced two more Lyapunov functions which cannot be derived from the IQC theorem: One is in continuous time (Theorem 3.2), and the other one is in discrete time (equation (7)). The latter is the first Lyapunov function that uniformly captures the discrete time behavior of the algorithms for different choices of parameters.
> - **Use of well-known techniques**: It is true that we use well-established techniques like Lyapunov analysis and Grönwall lemma. Nevertheless, we leverage these tools to achieve a family of Lyapunov functions whose form is not seen before in the literature. Crucially, our analysis is NOT a trivial extension of another work, the form of the unified Lyapunov function requires different steps in the analysis. For example, we use a limit analysis after (55) in the appendix which is different from all of the prior works. This is just one example of the many analytical and algebraic complexities that arise from the generality of the Lyapunov function we addressed in our analysis.
> - **Improvements are in the constants**: We agree that the rates are linear similar to the state-of-the-art. However, it is important to note that the improvements are not in the constants but in the exponents of the linear rate, which significantly tightens the bounds.
> For example, Theorem 4.1 proves a convergence rate of order $C_{GM}(1-\sqrt{ 1/\kappa})^k)$, and the best previous result in this setting was $5L||x_0-x^*||^2_2(1+\sqrt{ 1/\kappa}/9)^{-k})$. Suppose $L=10,\mu=1,\kappa=10$ and $k=50$, then $C_{GM}(1-\sqrt{1/\kappa})^k \approx 2.35\times 10^{-7}\times ||x_0-x^*||^2_2$ while $5L\|x_0-x^*\|^2_2((1+\sqrt{ 1/\kappa}/9)^{-k}) \approx 8.8939||x_0-x^*||_2^2$.
> Similarly, for QHM, we prove a rate of order $O((1-\sqrt{\frac{1}{3\kappa}})^k)$ while the previous best rate was $O((1+\frac{1}{40\sqrt{\kappa}})^{-k})$. If $\kappa=10$ and $k=50$, then $(1-\sqrt{\frac{1}{3\kappa}})^k \approx 4.19 \cdot 10^{-5}$ while $(1+\frac{1}{40\sqrt{\kappa}})^{-k} \approx 0.6745$.
> To highlight our improvements, we have now added Table 2 to showcase the faster convergence rates both in continuous and discrete-time.
>
> - **Comparison with GM-ODE**: We discussed the differences between GM-ODE and GM2-ODE in detail in pages 8-9 in our submission. The main drawback of GM-ODE is its inconsistency in terms of ODE and algorithm recovery. Moreover, GM2-ODE is more capable of recovering algorithms and leads to better convergence guarantees. Please note that the Lyapunov function corresponding to GM-ODE does not achieve the optimal rates. Also note that discretization of GM-ODE using SIE method does not exactly recover the NAG algorithm. Finally, we would like to draw your attention to the ability of GM2-ODE to recover other high-resolution ODEs.
> Our unified model (GM^2-ODE) has the following two attractive features:
> 1. Not only does it unify the ODEs in continuous time, but also existing algorithms through SIE discretization. Surprisingly, all these algorithms are found to be well-known methods (NAG,HB,TMM, etc) and they are all recovered through the same routine.
>  2. The corresponding *unifying* Lyapunov function leads to better convergence rates than the previous results obtained by continuous time analysis. To further emphasize this, we added Table 2 to compare our rates with existing results, which shows precisely the improvements.
> - **Less condition on the convergence of the QHM algorithm**: Please note that the previous rate on QHM requires $L/\mu\geq 9$ [6], while our result in corollary 4.1.1 is free from such condition.
> - **Does GM2-ODE lead to new methods?**: As shown in Table1, (GM2-ODE) and its SIE discretization recover the TM method and its ODE through different choices of coefficients. In addition, the current existing rates on (HR_TM) in the paper are not comparable to the algorithm's convergence rate ($\mathcal O\left((1-\sqrt{\tfrac{\mu}{L}}\right)^{2k}$ which should correspond to $\mathcal{O}\left(e^{-2\sqrt{\mu}t}\right)$ in continuous time). This observation suggests a new high-resolution ODE for the TM method. Setting the same coefficients that recover the TM method in (GM2-ODE) reads
>
> $$\ddot{X}_t+\sqrt{\mu}\left(\frac{3-\sqrt{\tfrac{\mu}{L}}}{1-\sqrt{\tfrac{\mu}{L}}}\right)\dot X_t +\frac{1}{\sqrt{L}}\nabla^2 f(X_t)\dot X_t+\left(\frac{2}{1-\sqrt{\tfrac{\mu}{L}}}+\sqrt{\frac{\mu}{L}}\right)\nabla f(X_t)=0.$$
>
> Surprisingly, if the step-size $\tfrac{1}{\sqrt{L}}\rightarrow 0$ the above ODE reduces to $$ \ddot{X}_t+3\sqrt{\mu}\dot X_t+2\nabla f(X_t)=0$$
> which is the low-resolution ODE corresponding to the TM method [3]. This discussion will be added to the paper.

---

> > ### Author Response · Authors · 2023-11-20
> >
> > # Presentation
> > - **On the Abstract**: Thank you for bringing this up. We will revise our abstract to be more specific about the strongly convex problem setting, discretization techniques used, and the improvements achieved in the rate.
> > - **On the Introduction**: We believe that the first two paragraphs provide insight to the reader as to why the phenomenon of acceleration in discrete time analysis is considered mysterious and explain why other approaches, including continuous-time analysis, are studied for these methods. But we can shorten these two sections to save space and extend our discussions on the continuous-time perspective and why we study this specific perspective.
> > - **Missing smoothness assumption in Thm 3.1**: We would like to thank you for your careful reading of our paper, including the appendix. However, while there is a smoothness assumption in Theorem A.1, we intentionally discard it in Theorem 3.1. Please note that the smoothness assumption is crucial for the discrete-time analysis of first-order methods, but its effect is negligible in the continuous-time analysis, as stated in [1, the last sentence in the introduction]. The coefficient $\sigma$ is defined as a nonnegative constant of the matrix $M^{(3)},$ which is the only matrix that relies on the smoothness assumption. Therefore, $\sigma$ is defined in Theorem A.1, and we explicitly set $\sigma=0$ in Appendix B.2 below equation (31). This choice effectively removes any influence of the smoothness constant $L$ and the smoothness assumption from the theorem.
> > - **Unclear notations**: The curly $F$ notation denotes the set of $\mu$-strongly convex and $L$-smooth functions. We will clarify this in the revision to avoid ambiguity. Other minor comments and typos will be fixed.
> >
> > # Answers to your Questions
> > 1. The Semi-Implicit Euler (SIE) is a symplectic discretizer, meaning that it preserves the energy of Hamiltonian systems after discretization. Therefore, it was used in previous work to show the acceleration of the NAG algorithm [5]. Being the simplest symplectic discretization, it is well-suited for our analysis, as you pointed out, ensuring consistency in our work. This means that after discretization, we recover algorithms using the same coefficients in the corresponding ODE. A more detailed discussion on this subject can be found in subsection 5.2 in our submission.
> > 2. They also use the SIE discretizer. This is mentioned in Step3 of their paper page 11. We will clarify this in the paper.
> > 3. Thank you for bringing up this work. It is mentioned in [2], appendix A.3, end of paragraph 1, that for the case of NAG and HB, applying their analysis does not yield novel interpretable insights on the differences between the two methods. We also believe no difference would be gained by applying their form of discretization to our problem.
> > 4. Yes, one can apply a similar approach to the convex setting. In fact, in a concurrent work [4], a similar model is investigated for the convex (and also for stochastic) setting. Specifically, by taking $\alpha_t=\log(n’), \beta_t=q’t, \gamma_t=n’t$ in the Strongly Convex force in [4], we can find $n=n’,q=q’,m=\sqrt{s}n’^2/\mu,p=n’/2$. These parameters are found under the convergence conditions of Theorem 2.3 in [4] and are aligned with the convergence conditions of Theorem 3.2 in our paper.
> > 5. The known rate constant is $(mu+L)/2$ for the NAG. Our constant for the case of NAG reads
> > $\frac{L\kappa}{2} (1-\sqrt(1/\kappa)) + \frac{L}{2}(1-\sqrt(1/\kappa)/2)$.
> >
> > # References
> > [1] J.M. Sanz-Serna and K.C. Zygalakis. "The Connections Between Lyapunov Functions for Some Optimization Algorithms and Differential Equations." SIAM Journal on Numerical Analysis, 59(3):1542–1565, 2021.
> >
> > [2] J. Chavdarova, et al. "Last-Iterate Convergence of Saddle-Point Optimizers via High-Resolution Differential Equations." 2023.
> >
> > [3] J. Kim and I. Yang. "Convergence Analysis of ODE Models for Accelerated First-Order Methods via Positive Semidefinite Kernels." NeurIPS (2023).
> >
> > [4] H. Maskan, et al. "A Variational Perspective on High-Resolution ODEs." arXiv:2311.02002 (2023).
> >
> > [5] B. Shi et al. "Acceleration via Symplectic Discretization of High-Resolution Differential eqEquations." arXiv:1902.03694 (2019).
> >
> > [6] Zhang, Peiyuan et al. “Revisiting the Role of Euler Numerical Integration on Acceleration and Stability in Convex Optimization.” ArXiv abs/2102.11537 (2021).

---

### Meta-Review · Area_Chair_V8ko · 2023-12-02

**Metareview:**

This paper studies high-resolution ODE (HR-ODE) for understanding accelerated algorithms. The authors provided some unified HR-ODE that recovers some existing ones along this line of research. It is found that the contribution of the paper is incremental, as it doesn't lead to new algorithms with promising practical performance. The presentation also needs to be improved to better highlight the main contributions of the paper.

**Justification For Why Not Higher Score:**

The contribution is incremental. It can't be accepted in its current form.

**Justification For Why Not Lower Score:**

NA.

---

### Decision · Program_Chairs · 2024-01-16

Reject